# Prevalence of childfree adults before and after Dobbs v Jackson in Michigan (USA)

**Jennifer Watling Neal** *, **Zachary P. Neal**

Psychology Department, Michigan State University, East Lansing, MI, United States of America

* jneal@msu.edu

## Abstract

Childfree adults are the most common type of non-parent in the United States and are distinguished by their lack of desire to have children. Although there are many reasons one may choose not to have children, recent restrictions on reproductive health care may also contribute to this decision. For example, the United States Supreme Court's decision in *Dobbs v. Jackson* eliminated a long-standing constitutional protection for abortion access, which reduced patients' medical autonomy and increased the risks of pregnancy and childbirth, and therefore may have led adults to decide not to have children. In this study, we use representative data on Michigan adults immediately before and after the *Dobbs* decision to examine changes in the prevalence of childfree adults in this population. We find that 21% of Michigan adults were childfree before the *Dobbs* decision, but this number rose to nearly 26% after the decision. Controlling for demographic characteristics, a Michigan adult was 32.8% more likely to be childfree after the *Dobbs* decision than before. We conclude that when access to safe reproductive health care is uncertain or unavailable, adults that do not already have children may decide that they do not want children.

## Introduction

Childfree adults are the most common type of non-parent in the United States [1–4] and are distinguished by their lack of desire to have children [5]. Childfree adults' reasons for not wanting children are diverse and can include both individual and societal factors. At the individual-level, many simply report they do not want children [1], while others point to health concerns or a desire for personal advancement [6]. At the societal-level, greater gender equality, the broader political climate, and economic factors may be associated with not wanting children [7]. This study focuses on one important societal-level factor that might inform adults' decisions to be childfree: restrictions on access to reproductive health care.

In the United States, recent court rulings have paved the way for new restrictions on access to reproductive health care in general, and to abortion in particular. In June 2022, the US Supreme Court issued a decision in *Dobbs v. Jackson Women's Health Organization* (hereafter, *Dobbs*) [8]. The *Dobbs* decision overturned a federal constitutional protection to abortion that was established by the US Supreme Court's earlier decision in *Roe v. Wade* (hereafter, *Roe*) almost 50 years earlier [9]. This decision had almost immediate repercussions for access to reproductive health care. "Trigger laws" (i.e., new laws that would take effect if *Roe* were

**Data Availability Statement:** The data underlying the results presented in the study are available from https://osf.io/w6yzf/.

**Funding:** This work was supported in part by a Michigan Applied Public Policy Research award to

JWN and ZPN. The sponsors did not play a role in this work.

**Competing interests:** The authors have declared that no competing interests exist.

overturned) in thirteen states imposed bans or severe restrictions on abortions, while "zombie laws" (i.e., old laws restricting abortion that were never repealed after the passing of *Roe*) in several other states including Michigan [10] made the future of abortion access uncertain. In addition to abortion, the *Dobbs* decision created uncertainty about future access to other types of reproductive health care, including miscarriage management as well as temporary (i.e., birth control) and permanent (i.e., sterilization) forms of contraception.

Restrictions related to the *Dobbs* decision may have led more adults to decide they do not want children, and therefore to be childfree, for several reasons. For women, these restrictions meant both a loss of autonomy regarding reproductive decisions and increased health risks because certain forms of care would be unavailable or difficult to access. For men, they meant increased health risks to their partners. For both women and men, these restrictions may also signal broader trends toward authoritarianism, and therefore toward a world into which they would not want to bring children.

This study aims to explore whether the restrictions and uncertainties of the *Dobbs* decision are associated with changes in the prevalence of childfree adults. Specifically, we ask whether adults in Michigan were less likely to want children, and thus more likely to be childfree, after the *Dobbs* decision. We begin by reviewing the literature on childfree adults and explain why restrictions on reproductive rights may lead to higher prevalence rates. Then we describe the *Dobbs* decision within the US national context and the Michigan context and the implications of this decision for the prevalence of childfree adults in Michigan. Next, using representative samples of Michigan adults, we examine changes in the prevalence of childfree adults before and after the *Dobbs* decision. Finally, we conclude with implications for future research and for policy.

## Background

### Childfree adults in the US

*Childfree* adults, also known as voluntarily childless adults, do not have and do not want children [5]. Because childfree adults do not have children, they are distinctly different from *parents*. Additionally, because childfree adults actively do not want children, they are distinctly different from four other groups of non-parents who (may) want or (may have) wanted children [3–5, 11]. *Not-yet-parents* are planning to have children in the future, *undecided* non-parents may have children in the future but have not decided, *ambivalent* non-parents will not have children in the future and do not know if they would have wanted children, and *childless* non-parents wanted children but could not have them due to medical or social circumstances.

In the United States, nearly half of all non-parents are childfree [1]. Thus, childfree adults represent the largest group of non-parents. Longitudinal data from the National Survey of Family Growth (NSFG) indicated that the percent of women age 15 to 49 who are childfree has nearly doubled from 6% in 2010 to 9.8% in 2019 [12]. Similarly, data from the NSFG indicated that the percent of men age 15 to 49 who are childfree has doubled from 9.9% in 2012 to 20.2% in 2018, while data from the Monitoring the Future survey indicated that the percent of male high school seniors who are childfree more than tripled from 1.7% in 2000 to 6.2% in 2019 [13]. These findings point to a long-term trend of increasing numbers of childfree adults. Finally, recent work in Michigan suggests that the prevalence of childfree adults may be much higher than previously estimated, totaling 21.35% in data pooled from 2021 and 2022 [4].

The number of adults who are childfree is growing in the US and is quite large in Michigan [3, 4]. Therefore, it is important to understand factors that might be associated with the prevalence of childfree adults. Individual-level factors are often offered as a reason for being childfree including a desire for freedom, a focus on investing in existing interpersonal and

romantic relationships, career investment, health concerns, or a general dislike of children [6, 7, 14, 15]. However, societal-level factors including greater gender equality, the broader political climate, and economic factors may also lead adults to be childfree [7, 16, 17]. Recent political and legal changes regarding reproductive health care may represent an additional reason that both women and men choose to be childfree. For women, restrictions on access to reproductive health care represent a limit on medical autonomy, and can introduce health risks by blocking access to medically necessary procedures such as abortions for non-viable pregnancies. Although men's health may not be directly impacted, such restrictions may mean increased health risks to their partners, leading them to re-evaluate their own intentions to have children. Moreover, for both women and men, politically-motivated restrictions on reproductive rights may signal worrying trends toward authoritarianism, leading to concerns about what the future could hold for any children they might have.

## The Dobbs decision

In the US, laws regulating abortion have a complex history and access to reproductive health care has varied over time. Before 1973, access to abortion was regulated by state governments. In most states, abortion was legal only under limited circumstances, leading many of their residents to seek termination of their pregnancies through underground social networks [18]. In a 1973 7-2 ruling, the landmark US Supreme Court decision in *Roe* established a federal constitutional protection to an abortion during the first trimester, after which restrictions were permitted if a state could demonstrate a compelling need for such regulation. Subsequent rulings further refined the federal regulation of abortion. However, they each preserved the essential holding of *Roe*, that pregnant people have a constitutional right to an abortion under the due process clause of the 14[th] amendment to the United States Constitution.

In May 2022, a 98-page draft US Supreme Court opinion written by Justice Samuel Alito was leaked to the media [19]. This draft opinion, written for the *Dobbs* case, signaled that the court was planning to overturn *Roe* and sparked fears about future access to reproductive health care in the US. In June 2022, these fears were realized when the final *Dobbs* decision was issued in a narrow 5-4 ruling and abruptly ended the constitutional protection for abortion access provided by *Roe* [8]. In the majority opinion for *Dobbs*, Alito argued that "The Court finds that the right to abortion is not deeply rooted in the Nation's history and tradition," while in a concurring opinion, Justice Clarence Thomas suggested that the Supreme Court may reconsider other reproductive rights including a right to contraceptive use.

The *Dobbs* decision had immediate consequences for US citizens' access to reproductive health care. In thirteen states, trigger laws were activated that criminalized or severely restricted abortions. These laws made it difficult to obtain abortion services, but also complicated the provision of medical and surgical treatments for miscarriage [20, 21]. In several other states, old "zombie" laws that had outlawed abortion prior to *Roe* and were never repealed created legal chaos and uncertainty about access to reproductive rights. Even in states where abortion was legal and/or constitutionally protected, the threat of a federal abortion ban and the possibility that the US Supreme Court might reconsider federally protected rights to contraceptives created an air of unease.

Access to reproductive health care following *Dobbs* was particularly uncertain in the state of Michigan, where a complex set of laws and court rulings created significant ambiguity (see Table 1). Before *Roe*, a 1931 law criminalized abortion except when a pregnant person's life was in danger [10]. This "zombie" law was not repealed after *Roe*, and therefore would resume its legal effect in the event that *Roe* was ever overturned. Anticipating the possibility that *Roe* would be overturned, in early 2022 both Planned Parenthood and Michigan Governor

**Table 1. Selected legal milestones in Michigan reproductive rights.**

| Date | Law / Case | Effect |
|---|---|---|
| Sep 18, 1931 | Michigan Penal Code §750 | Criminalized abortion |
| Jan 22, 1973 | *Roe v. Wade* | Legalized abortion; Invalidated MPC§750 |
| May 17, 2022 | *Planned Parenthood v. Attorney General* | Temporarily prevented enforcement of MPC§750 if *Roe* was overturned |
| Jun 24, 2022 | *Dobbs v. Jackson* | Overturned *Roe* |
| Aug 1, 2022 | *In re Jarzynka* | Criminalized abortion; Allowed county prosecutors to enforce MPC§750 |
| Aug 1, 2022 | *Whitmer v. Linderman et al.* | Legalized abortion; Temporarily prevented county prosecutors from enforcing MPC§750 |
| Sep 7, 2022 | *Planned Parenthood v. Attorney General* | Permanently prevented enforcement of MPC§750 |
| Nov 8, 2022 | Proposition 3 | 56.7% of Michigan voters approved of 'Right to Reproductive Freedom' Amendment |
| Dec 24, 2022 | Michigan Constitution, Art. 1, §28 | Amended Michigan constitution declaring that "Every individual has a fundamental right to reproductive freedom" |
| Apr 5, 2023 | Act 11 of 2023 | Repealed MPC§750 |

Gretchen Whitmer filed preemptive lawsuits to block its enforcement. A temporary injunction was granted in May, preventing the law's enforcement in the event *Roe* was overturned. This temporary injunction blocked the *Dobbs* decision from having a legal impact in Michigan when it was handed down in June. The following week a different court ruled that the injunction did not apply to county prosecutors, allowing them to enforce the 1931 law and effectively re-criminalizing abortion in Michigan. However, later the same day still a third court issued a temporary restraining order blocking the law's enforcement by county prosecutors, effectively re-legalizing abortion in Michigan.

Enforcement of the 1931 law was permanently blocked in September, and was rendered unconstitutional in November when 56.7% of Michigan voters approved the "Right to Reproductive Freedom" amendment. This amendment constitutionally protected Michigan residents' rights to "effectuate decisions about all matters relating to pregnancy, including but not limited to prenatal care, childbirth, postpartum care, contraception, sterilization, abortion care, miscarriage management, and infertility care" (Article I, §28). The 1931 law was rendered unconstitutional by the amendment, and was formally repealed by the Michigan legislature in Act 11 of 2023. Although Michigan residents retained access to reproductive health care throughout this process, because each case could have had a different outcome and additional restrictions could have been introduced by the Republican-controlled state legislature, the process nonetheless introduced significant uncertainty and anxiety around these issues [22, 23].

## The current study

The *Dobbs* decision overturned the nearly 50-year constitutional right to an abortion set by *Roe*, creating significant barriers to reproductive health care for millions of Americans and great unease about the future of reproductive health care for all Americans. *Dobbs* not only limited abortion access, but also limited medical and surgical miscarriage management, and led to questions about future access to contraceptives. In Michigan, residents experienced a roller coaster of uncertainty about their access to reproductive health care due to a "zombie"

law, a series of lawsuits and rulings in state courts, the passage of Proposition 3, and subsequent amendment to the state's constitution. These changes in access to reproductive health care represent a societal-level factor that could affect whether Michigan adults want children. Specifically, these changes impact women directly by limiting their health care options, impact men indirectly by limiting their partners' health care options directly, and impact women's and men's views of the future political climate.

In this study, we explore the prevalence of childfree adults in Michigan before and after the *Dobbs* decision using data from representative cross-sectional samples collected as part of the Michigan State of the State Survey (SOSS). These data were collected during two periods shortly before the *Dobbs* leak (September 2021 and April 2022) and two periods shortly after the *Dobbs* decision (September 2022 and December 2022). The availability of comparable data before and after *Dobbs* provided a unique opportunity to examine whether changes in reproductive health care access represent a societal-level factor associated with population trends in childfree prevalence. We use these data to answer two research questions. First, did the percent of Michigan adults who are childfree increase following *Dobbs* (Research Question 1)? Second, was a Michigan adult more likely to be childfree following *Dobbs*, controlling for their demographic characteristics (Research Question 2).

## Methods

### Samples

Our data come from the Michigan State of the State Survey (SOSS), which is collected by You-Gov under contract to the Institute for Public Policy and Social Research at Michigan State University. The SOSS is conducted several times throughout the year with a consistent set of demographic questions, and a rotating set of researcher-commissioned questions. Each wave is collected from a sample of 1000 Michigan adults ranging in age from 18 to 96, and is weighted to make the sample representative of the Michigan adult population with respect to gender, age, race, and education. In this study, we use four cross-sectional waves of the SOSS (see Table 2). The Michigan State University Institutional Review Board determined this study to be 'not human subjects' on 25 April 2023 (STUDY00009134) because the data are existing, public, and de-identified.

### Measures

**Childfree adults and other family statuses.** Respondents' family status was measured using a three-question funnel design [5] that has previously been used to estimate the

**Table 2. State of the state survey data.**

| Period | SOSS Wave | Date(s) | Sample Size | |
|---|---|---|---|---|
| | | | **Full** | **Complete** |
| Pre-*Dobbs* | 82 | September 3-27, 2021 | 1500[a] | 1458 |
| | 84 | April 12-21, 2022 | 1000 | 979 |
| *Dobbs* leak | – | May 2, 2022 | | |
| *Dobbs* decision | – | June 24, 2022 | | |
| Post-*Dobbs* | 85 | September 2-15, 2022 | 1000 | 976 |
| | 86 | December 9-19, 2022 | 1000 | 989 |

[a] This wave included an oversample (*N* = 500) of parents.

prevalence of childfree adults [3, 4, 11]. Respondents are first asked if they have ever had children. Those who answer 'yes' are *parents*, while those who answer 'no' are non-parents. To clarify their fertility intentions, non-parents are then asked: "Do you plan to have children in the future?" Those who answer 'yes' are *not-yet-parents*, those who answer 'don't know' are *undecided*. Those who answer 'no' are asked a final question: "Do you wish you had or could have had children?" Those who answer 'yes' are *childless*, those who answer 'don't know' are *ambivalent*, and those who answer 'no' are *childfree*. Thus, respondents are classified as childfree if they neither have nor want children. Across all waves, 79 (1.8% of the full sample) respondents failed to answer one or more of these questions and therefore have an unknown family status.

**Demographics.** Respondents were asked "What is your sex?", to which they could respond male, female, or intersex/other. We measure sex using a binary variable coded 1 for males, and 0 for female. In wave 82, two respondents failed to report their sex and seven respondents reported being intersex. These nine cases (0.2% of the full sample) are treated as missing on the sex variable.

Respondents were asked "In what year were you born?", from which their age can be computed by subtracting the survey's year from their response. We measure age using a binary variable, where 1 indicates age 45 or older, and 0 indicates age 44 or younger. We use a binary variable to facilitate reporting prevalence and testing differences using a t-test, however the results remain the same if a continuous measure of age is used instead.

Respondents were asked "What is the highest level of education you have completed?", and provided a response from a 10-point ordinal scale ranging from 'did not go to school' to 'graduate degree.' We measure education using a binary variable, where 1 indicates completion of a 4-year college degree or higher and 0 indicates no college degree. We use a binary variable to facilitate reporting prevalence and testing differences using a t-test, however the results remain the same if an ordinal measure of education is used instead.

Respondents were asked "Are you currently married, divorced, separated, widowed, a member of an unmarried couple, or have you never been married?" We measure relationship status using a binary variable, where 1 indicates the respondent is currently partnered (married, remarried, or member of an unmarried couple) and 0 indicates the respondent is not currently partnered (divorced, separated, widowed, or single). Across all waves, 10 (0.2% of the full sample) failed to provide a response to this question.

Finally, respondents were asked "What is your race?", and were permitted to indicate whether they were a member of each of 6 racial groups. Because the number of respondents reporting membership in each non-White racial was relatively small, we measure race using a binary variable, where 1 indicates the respondent is White alone, and 0 indicates the respondent reported membership in one or more non-White racial groups.

## Analysis

As noted above, a small number of cases (*N* = 98, 2.2% of the full sample) are missing observations of family status, sex, or relationship status. In all analyses, these cases are dropped listwise, yielding an analytic sample of 4402 cases.

Because our research questions are focused on differences before and after the *Dobbs* decision, to maximize the available sample and statistical power, we pool data collected in September 2021 (wave 82, *N* = 1458) and April 2022 (wave 84, *N* = 979) to represent the pre-*Dobbs* period, and pool data collected in September 2022 (wave 85, *N* = 976) and December 2022 (wave 86, *N* = 989) to represent the post-*Dobbs* period. Preliminary analyses confirm that these samples are sufficiently homogeneous to warrant pooling. Specifically, the prevalence of

childfree adults in the two pre-*Dobbs* waves (wave 82 = 21.2%; wave 84 = 20.9%) is not statistically significantly different ($t$[2435] = −0.102, $p$ = 0.92). Likewise, the prevalence of childfree adults in the two post-*Dobbs* waves (wave 85 = 25.9%; wave 86 = 26.0%) is not statistically significantly different ($t$[1963] = 0.038, $p$ = 0.97). More broadly, the prevalence of each family status is stable within the two pre-*Dobbs* waves, and within the two post-*Dobbs* waves. After pooling the data, we adjusted the sampling weights to account for differing sample sizes by wave [24].

To evaluate Research Question 1—did the percent of Michigan adults who are childfree increase following *Dobbs*—we use a t-test to compare the prevalence of childfree adults before and after *Dobbs*. To evaluate Research Question 2—was a Michigan adult more likely to be childfree following *Dobbs*, controlling for their demographic characteristics—we estimate a logistic regression predicting whether an adult is childfree as a function of time (i.e. pre- or post-*Dobbs*), controlling for their sex, age, education, relationship status, and race. For both tests, we use the `R survey` package [25] to incorporate sampling weights. The data and code necessary to replicate the results reported below are available at https://osf.io/w6yzf/.

## Results

Table 3 reports the prevalence of family status and demographic groups before and after the *Dobbs* decision. The first row provides an answer to Research Question 1: We find that the percent of adults who are childfree statistically significantly increased, from 21.11% before *Dobbs*, to 25.92% after *Dobbs* ($t$[4400] = 2.88, $p < 0.01$). The middle panel of Table 3 shows the change in prevalence of other family statuses. Because an increase in the prevalence of childfree adults must be accompanied by a decrease in the prevalence of other family statuses, these values provide some insight into why we observe an increase in the prevalence of childfree adults. We observe that while the prevalence of childfree adults increased, the prevalence of both not-yet-parents (from 10.24% to 7.43%) and childless (from 5.31% to 2.98%) adults significantly decreased after *Dobbs*. The bottom panel of Table 3 shows that because the demographic composition of the Michigan adult population did not change over this short period, as expected, the prevalence of demographic characteristics are not statistically significantly different before and after *Dobbs*.

**Table 3. Prevalence of family status and demographic groups before and after the *Dobbs* decision.**

| Group | Pre-*Dobbs* | | Post-*Dobbs* | | T-test |
|---|---|---|---|---|---|
| | Percent | SE | Percent | SE | |
| Childfree | 21.11 | 1.22 | 25.92 | 1.14 | t[4400] = 2.88, p < 0.01 |
| Other family statuses | | | | | |
| Parent | 50.87 | 1.39 | 51.88 | 1.31 | t[4400] = 0.53, p = 0.60 |
| Not yet parent | 10.24 | 1.02 | 7.43 | 0.73 | t[4400] = -2.24, p = 0.03 |
| Undecided | 9.05 | 0.87 | 9.26 | 0.82 | t[4400] = 0.18, p = 0.86 |
| Childless | 5.31 | 0.65 | 2.98 | 0.42 | t[4400] = -3.01, p < 0.01 |
| Ambivalent | 3.42 | 0.53 | 2.53 | 0.39 | t[4400] = -1.34, p = 0.18 |
| Demographics | | | | | |
| Male | 47.01 | 1.4 | 49.01 | 1.32 | t[4400] = 1.04, p = 0.3 |
| Age 45+ | 57.61 | 1.4 | 56.78 | 1.33 | t[4400] = -0.43, p = 0.67 |
| College Degree | 27.86 | 1.11 | 26.86 | 1.06 | t[4400] = -0.65, p = 0.52 |
| Partnered | 51.36 | 1.39 | 54.28 | 1.32 | t[4400] = 1.52, p = 0.13 |
| White | 81.73 | 1.17 | 81.57 | 1.08 | t[4400] = -0.1, p = 0.92 |

**Table 4. Logistic regression predicting whether an adult ($N$ = 4402) is childfree.**

| | Without covariates | | | | With covariates | | | |
|---|---|---|---|---|---|---|---|---|
| | **B** | **SE** | **P** | **OR** | **B** | **SE** | **P** | **OR** |
| Intercept | -1.318 | 0.073 | < 0.001 | 0.268 | -1.582 | 0.146 | < 0.001 | 0.206 |
| Post-*Dobbs* | 0.268 | 0.094 | 0.004 | 1.307 | 0.284 | 0.095 | 0.003 | 1.328 |
| Male | | | | | 0.411 | 0.097 | < 0.001 | 1.508 |
| Age 45+ | | | | | -0.033 | 0.105 | 0.756 | 0.968 |
| College Degree | | | | | 0.043 | 0.099 | 0.664 | 1.044 |
| Partnered | | | | | -0.565 | 0.106 | < 0.001 | 0.568 |
| White | | | | | 0.403 | 0.146 | 0.006 | 1.496 |

Table 4 reports two logistic regressions predicting whether a Michigan adult is childfree. The coefficients in the model without covariates exactly reproduce the prevalence values reported in Table 3 (e.g., $\frac{e^{-1.318}}{1+e^{-1.318}} = 0.2111$). The model with covariates provides an answer to research question 2: We find that, after controlling for demographic characteristics, an adult was statistically significantly more likely to be childfree after *Dobbs* ($B$ = 0.284, $SE$ = 0.095, $p$ = 0.003). The odds ratio indicates that, after controlling for demographic characteristics, an adult was 32.8% more likely to be childfree after *Dobbs*.

## Discussion

When the US Supreme Court issued the *Dobbs* decision overturning a federal constitutional right to an abortion, it had swift effects on Americans' access to reproductive health care, leading to bans and restrictions in some states and fears about continued access in others. In Michigan, a 1931 "zombie law" and subsequent lawsuits and rulings created uncertainties around residents' access to reproductive health care. In this paper, we aimed to understand whether these uncertainties are associated with changes in the prevalence of childfree adults. In particular, we examined whether the percent of Michigan adults who are childfree increased following the *Dobbs* decision. Additionally, we examined whether a Michigan adult was more likely to be as childfree after *Dobbs*, controlling for common demographic characteristics.

The *Dobbs* decision and subsequent uncertainties in Michigan may have resulted in more adults deciding they never want to have children, that is, deciding to be childfree. Specifically, the prevalence of Michigan adults who were childfree increased by 4.8 percentage points in a relatively short timeframe before and after the *Dobbs* decision. Controlling for common demographic factors, a Michigan adult was 32.8% more likely to be childfree after the *Dobbs* decision than before it. This is a fairly large effect and suggests that the *Dobbs* decision may have dampened many Michigan adults' interest in having children.

There are multiple potential explanations for where these additional childfree adults came from. One possibility is that many childfree adults moved to Michigan, and many non-childfree adults moved out of Michigan, between these two time points. While such shifts in the state's population could generate the changes in prevalence that we observe, it is unlikely that such a large and systematic migration occurred over such a short period of time. Instead, it is more likely that Michigan adults who were not childfree before *Dobbs* became childfree after *Dobbs*. The significant declines in prevalence for both childless adults and not-yet-parents shown in Table 3 point to some possible family status transitions. It is possible that childless individuals who had wanted (but could not have) children before *Dobbs* decided, following *Dobbs*, that they in fact did not want children. Likewise, it is possible that not-yet-parents who

were planning to have children before *Dobbs* decided, following *Dobbs*, that they no longer wanted to have children. If substantial numbers of non-yet-parents became childfree following, and possibly because of *Dobbs*, this would have significant implications for demographic trends in, and policies intended to boost, fertility rates. This could also help explain the increase in patients' requests for permanent methods of contraception like vasectomy, bilateral salpingectomy, and tubal ligation [26, 27]. However, because these data are anonymous and cross-sectional, we are unable to explicitly test these potential explanations, and specifically are unable to observe within-person changes in family status.

Given the observational nature of this study, we cannot definitively rule out other potential explanations for changes in the population-level prevalence of childfree Michigan adults such as the COVID-19 pandemic, inflation, or climate change. However, our preliminary analyses and prior research suggest that the prevalence of specific family statuses were stable during the pre-*Dobbs* period when the COVID-19 pandemic, inflation, and climate change were already present but reproductive rights were still constitutionally protected [4]. Furthermore, our preliminary analyses also revealed that the prevalence of specific family statuses were stable during the post-*Dobbs* period, with the changes occurring specifically during the time when the *Dobbs* decision was issued.

This study has several strengths, including the use representative data from Michigan collected at multiple times before and after the *Dobbs* decision. However, these results should be interpreted in light of some limitations. First, this study relies on trend, not panel, data. Because the SOSS does not track the same individuals over time, we can draw conclusions about changes in the prevalence of childfree adults and other family statuses in the population, but are unable to draw conclusions about specific individuals' transitions between family statuses. Our findings would be strengthened by future work that uses panel data to track how restrictions on reproductive health care may lead individuals to change their decision about whether or not to have children. Second, this study relies on data from only one US state (Michigan). However, it was a state that experienced great uncertainty regarding access to reproductive health care after *Dobbs*, but where residents also maintained legal access to abortion. It would be interesting to determine whether our findings generalize to states where abortion was banned by "trigger laws" (e.g., Texas) or to states where abortion was legally protected prior to the *Dobbs* decision (e.g., New Jersey). Finally, our study only examined changes in the prevalence of childfree adults immediately following the *Dobbs* decision. Future research is needed to understand if the changes we observe persist, particularly in states like Michigan where reproductive health care was subsequently protected by a constitutional amendment.

Changes in access to reproductive health care created by the *Dobbs* decision may have unintended consequences for individuals' decisions whether or not to have children. Specifically, when access to safe reproductive health care is uncertain or unavailable, adults that do not already have children may decide that they do not want children. As Americans continue to experience restrictions and uncertainty in access to reproductive health care, it will be important to continue to examine how these changes impact adults decisions about whether to have children, and the impact that such decisions have on demographic trends and reproductive health policy.

## Author Contributions

**Conceptualization:** Jennifer Watling Neal, Zachary P. Neal.

**Data curation:** Jennifer Watling Neal, Zachary P. Neal.

**Formal analysis:** Jennifer Watling Neal, Zachary P. Neal.

**Funding acquisition:** Jennifer Watling Neal, Zachary P. Neal.

**Investigation:** Jennifer Watling Neal, Zachary P. Neal.

**Methodology:** Jennifer Watling Neal, Zachary P. Neal.

**Writing – original draft:** Jennifer Watling Neal, Zachary P. Neal.

**Writing – review & editing:** Jennifer Watling Neal, Zachary P. Neal.

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
