## [Decision Letter · Decision Letter 0]

25 Sep 2023

PONE-D-23-23066Fertility intentions among non-parents in Michigan (USA) before and after Dobbs v. JacksonPLOS ONE

Dear Dr. Watling Neal,

Thank you for submitting your manuscript to PLOS ONE. After careful consideration, we feel that it has merit but does not fully meet PLOS ONE’s publication criteria as it currently stands. Therefore, we invite you to submit a revised version of the manuscript that addresses the points raised during the review process. The article needs some minor formal revisions, in particular the methods and discussions sections needs a deepening, as indicated by the reviewers.

We look forward to receiving your revised manuscript.

Kind regards,

Andrea Cioffi

Academic Editor

PLOS ONE

Reviewers' comments:

Reviewer's Responses to Questions

**Comments to the Author**

1. Is the manuscript technically sound, and do the data support the conclusions?

Reviewer #1: Yes

Reviewer #2: Yes

2. Has the statistical analysis been performed appropriately and rigorously? 

Reviewer #1: I Don't Know

Reviewer #2: Yes

3. Have the authors made all data underlying the findings in their manuscript fully available?

Reviewer #1: Yes

Reviewer #2: Yes

4. Is the manuscript presented in an intelligible fashion and written in standard English?

Reviewer #1: Yes

Reviewer #2: Yes

5. Review Comments to the Author

Reviewer #1: This paper uses 4 waves of cross-sectional data from Michigan to investigate differences in fertility intentions among non-parents before and after the Dobbs decision. The findings show that the percent of non-parents that are categorized as childfree increased after Dobbs and that non-parents were more likely to be childfree (i.e., not want to have a child in the future) after Dobbs. Overall, the paper is well-written, clear, and addresses and important and timely issue. At the same time, there are some issues that need to be addressed. The lack of the analysis section and therefore the lack of information about the comparison category is why I answered "don't know" to the question about statistical analyses.

1. The authors need to add an analysis sub-section to the methods section. This sub-section should detail the methods that are used and clarify the comparison categories. For example, in the logistic regression models, the comparison category for childfree adults is not specified. I would assume that it is all other categories of non-parents, but this needs to be explained up front. Also, why use a logistic regression model rather than a multinomial logit model that can show differences across all the non-parent categories? This choice also needs to be explained and justified.

2. In the discussion, there is a bit of slippage in the language. Given that the authors are not using panel data (which is noted later in that section), the language needs to be clear in terms of comparing non-parents before and after Dobbs, rather than looking at interpersonal change over time. Along these lines, I find the sentence on page 7, line 268-269 a bit misleading. Additionally, the comparison category is never noted in the discussion of the results. In the sentence above, it needs to be clarified that this group (childfree individuals) is being compared to other non-parents, some of whom also do not want children, others are ambivalent, etc. The nuance is important for accurately portraying the results.

3. Relatedly, I would like to see more discussion of the other categories of non-parents. They are described in the beginning of the background section, but rarely mentioned after that. This shift in the text from discussing the categories to zeroing in on the childfree respondents is abrupt and unexplained. Please clarify which prevalence estimates you are referring to on page 2, lines 61 and 63. Is it all non-parents or those who are categorized as childfree?

4. Why would non-parents be more likely to be categorized as childfree after abortion was no longer federally protected? This is postulated throughout the paper in several places (pg. 3, line 80-81; pg. 5 line 153-154; pg. 7, lines 277-281), but no explanations are offered. I understand that you are not investigating that question in this paper, but offering some possible explanations for why this might be the case is important for justifying the hypotheses. In the discussion, it is mentioned that use of permanent methods of contraception have increased in the post-Dobbs period (pg. 8, line 328-9) and it is suggested that this supports the study’s results; however, increases in permanent contraception do not necessarily indicate changes in fertility intentions. Rather, these increases indicate changes in the use of these methods. That is, fertility intentions may have stayed the same, but childfree individuals may have sought out these permanent methods more frequently in order to ensure they remain childfree. Given that the paper’s argument rests on the connection between abortion access and fertility intentions, the links between access and intentions deserve more attention.

Reviewer #2: The present paper studies the impact of changes in laws pertaining to reproductive autonomy on people's decision to remain child free. In the context of Michigan, the authors find that after controlling for socio-demographic traits non-parents in Michigan were 62% more likely to not want children after the Dobbs decision than before it.

The paper is very well written and can be accepted as is for publication. One clarificatory question:

what is the age group considered for the study?

6. PLOS authors have the option to publish the peer review history of their article (what does this mean?). If published, this will include your full peer review and any attached files.

Reviewer #1: No

Reviewer #2: No

---

## [Decision Letter · Decision Letter 1]

2 Nov 2023

Prevalence of childfree adults before and after Dobbs v Jackson in Michigan (USA)

PONE-D-23-23066R1

Dear Dr. Neal,

We’re pleased to inform you that your manuscript has been judged scientifically suitable for publication and will be formally accepted for publication once it meets all outstanding technical requirements.

Kind regards,

Andrea Cioffi

Academic Editor

PLOS ONE

Additional Editor Comments (optional):

No further revisions are necessary.

Reviewers' comments:

Reviewer's Responses to Questions

**Comments to the Author**

1. If the authors have adequately addressed your comments raised in a previous round of review and you feel that this manuscript is now acceptable for publication, you may indicate that here to bypass the “Comments to the Author” section, enter your conflict of interest statement in the “Confidential to Editor” section, and submit your "Accept" recommendation.

Reviewer #1: All comments have been addressed

2. Is the manuscript technically sound, and do the data support the conclusions?

Reviewer #1: (No Response)

3. Has the statistical analysis been performed appropriately and rigorously? 

Reviewer #1: (No Response)

4. Have the authors made all data underlying the findings in their manuscript fully available?

Reviewer #1: (No Response)

5. Is the manuscript presented in an intelligible fashion and written in standard English?

Reviewer #1: (No Response)

6. Review Comments to the Author

Reviewer #1: (No Response)

7. PLOS authors have the option to publish the peer review history of their article (what does this mean?). If published, this will include your full peer review and any attached files.

Reviewer #1: No

---

## [Editor Report · Acceptance letter]

5 Nov 2023

PONE-D-23-23066R1 

Prevalence of childfree adults before and after Dobbs v Jackson in Michigan (USA) 

Dear Dr. Neal:

I'm pleased to inform you that your manuscript has been deemed suitable for publication in PLOS ONE. Congratulations! Your manuscript is now with our production department. 

Kind regards, 

on behalf of

Dr. Andrea Cioffi 

Academic Editor

PLOS ONE